mathematical modelling/health and disease and epidemiology

basic reproduction number, livestock trading, heterogeneity, endemic disease, generative modelling

**Author for correspondence:**
Martin A. Knight
e-mail: martin.knight@bioss.ac.uk

# Generative models of network dynamics provide insight into the effects of trade on endemic livestock disease

Martin A. Knight[1,2,3], Piran C. L. White[1], Michael R. Hutchings[3], Ross S. Davidson[2,3] and Glenn Marion[2]

[1]Department of Environment and Geography, University of York, Wentworth Way, York YO10 5NG, UK
[2]Biomathematics and Statistics Scotland, James Clerk Maxwell Building, Edinburgh EH9 3FD, UK
[3]Scotland's Rural College (SRUC), Peter Wilson Building, Edinburgh EH9 3JG, UK

MAK, 0000-0002-3812-4397

We develop and apply analytically tractable generative models of livestock movements at national scale. These go beyond current models through mechanistic modelling of heterogeneous trade partnership network dynamics and the trade events that occur on them. Linking resulting animal movements to disease transmission between farms yields analytical expressions for the basic reproduction number $R_0$. We show how these novel modelling tools enable systems approaches to disease control, using $R_0$ to explore impacts of changes in trading practices on between-farm prevalence levels. Using the Scottish cattle trade network as a case study, we show our approach captures critical complexities of real-world trade networks at the national scale for a broad range of endemic diseases. Changes in trading patterns that minimize disruption to business by maintaining in-flow of animals for each individual farm reduce $R_0$, with the largest reductions for diseases that are most challenging to eradicate. Incentivizing high-risk farms to adopt such changes exploits 'scale-free' properties of the system and is likely to be particularly effective in reducing national livestock disease burden and incursion risk. Encouragingly, gains made by such targeted modification of trade practices scale much more favourably than comparably targeted improvements to more commonly adopted farm-level biosecurity.

# 1. Introduction

The movement of animals via trade has long been considered a significant factor in the spread of disease within livestock populations [1–8]. For example, animal movements resulting from restocking following the 2001 foot-and-mouth disease (FMD) outbreak in Great Britain has been suggested as a contributing factor to the subsequent surge in bovine tuberculosis (bTB) positive farms [9,10]. The 2001 FMD outbreak itself spread widely, via animal movements [11], before detection led to national and international trade restrictions.

While exotic disease incursions like FMD in 2001 incur large costs over short timescales (estimates for FMD 2001 include up to UK £3.1 billion for stock losses [10] and £3.2 billion related to tourism [11]), many endemic diseases impact production year-on-year. For example, paratuberculosis (paraTB) reduces milk production in dairy cattle and causes weight loss affecting beef quality [12–14], and bovine viral diarrhoea virus (BVDV) often reduces fertility, animal growth and milk production [15]. These incur a significant cost to the agricultural industry (annually paraTB is estimated to cost £0.8 million, BVDV £39.6 million and bTB £29.7 million [16]). Unfortunately, controlling such diseases is a challenge due to a number of factors including animal movements, poorly understood transmission pathways (in particular the role of wildlife, e.g. rabbits and badgers in the spread of paraTB and bTB, respectively) [17–22], long latent periods [23] and variable sensitivities of diagnostic tests [24–26].

Understanding the initial spread of disease is highly informative of its long-term ability to persist within a system, and can be captured by each disease's basic reproduction number $R_0$; the number of secondary infections caused by a single infected individual in an otherwise susceptible population [27]. If $R_0 < 1$ then the disease is unable to persist and the disease-free critical point is stable. Conversely, if $R_0 > 1$, the disease-free critical point is unstable, and introduction of a small number of cases will result in exponential growth (initially) towards a critical point in which the disease persists. The stability of these critical points switch as $R_0$ passes through the threshold point $R_0 = 1$ [28]. Thus, sufficiently accurate models that retain analytical tractability so that expressions for $R_0$ can be obtained are of great value to inform effective interventions against both persistent disease and outbreaks.

The increasing availability of animal movement datasets has shed light on the complex and highly heterogeneous nature of livestock trade [29], with developments in network theory enabling new insights into the dynamics of such complex systems [30,31]. For example, the study of disease spread on such networks reveals that $R_0$ is heavily influenced by heterogeneity in the distribution of contacts [32,33]. Thus, to study the role of trade on disease spread, epidemiologists must develop models that adequately account for such complexities.

To date, attempts to assess the spread of disease in real-world cattle trade systems have largely consisted of replicating animal movements observed in data while overlaying simulated disease processes [5,7,34–36]. While these illustrate how past trade dynamics may have supported disease transmission, they cannot be generalized to ask 'what if …' questions about what might occur under some future set of trades. By contrast, generative models capable of capturing key properties of such systems, while not being restricted to replaying historic movements, would allow far more general conclusions to be drawn. They would enable exploration of the potential impact of changes in movement patterns, highlighting novel avenues for intervention and control that move beyond standard approaches based on improvements to on-farm biosecurity or movement standstills. Thus far, attempts to develop mechanistic generative models of livestock trade systems have focused on global properties [37] rather than considering trade between individual farms, or have modelled only the size and timing of animal movements on the frozen network of trade partnerships observed in the data [38].

To our knowledge, here we present the first truly generative mechanistic model for livestock trading systems. This accounts for heterogeneity between farms and stochastically generates both movement of animals between trade partners and dynamically evolves the underlying partnership network (§2.1). Extending this to account for disease transmission via trade, we apply and extend the results of [39] to account for between-farm heterogeneities and derive a per-farm $R_0$, denoted $R_0^i$ (§3.1). We subsequently use this analytic result to show large suppliers contribute disproportionately to disease spread and modifying trade dynamics could play a significant role in reducing disease burden (§3.2). With application to the Scottish cattle industry, we show that this parsimonious model can capture key features of the dynamics of a complex real-world trading system (§4). Subject to the condition that each farm maintains its annual in-flow of animals (representing maintenance of business requirements), we explore, for a broad spectrum of endemic diseases, the impact on $R_0$, the system-

average $R_0^i$, of changes to the way farms trade animals, including the formation of longer lasting trade partnerships. These results suggest that changes to trading practices are potentially effective in reducing both the burden of endemic disease and safeguarding against future disease outbreaks.

# 2. Material and methods

## 2.1. Livestock trading model

We seek to model *animal movements* in terms of *trading practices* consisting of the formation and cessation of trade partnerships and trading between established partners. Connectivity relevant to disease transmission (see §2.1.3) is therefore controlled by *partnership dynamics* (longevity of partnerships and number of concurrent partners) and *trading behaviour* (size and frequency of trades between partners). We assume a closed system of $N$ farms and summarize between-farm heterogeneity in terms of a small number of farm-level constants. Firstly, annual in- and out-flows of animals measure farm-level demand and supply for farm $i$, and are denoted by $\eta_i$ and $\zeta_i$, respectively. Secondly, rates quantifying the propensity for farm $i$ to form trading partnerships, $a_i$, end partnerships, $d_i$ and make trades, $b_i$. An outline of model quantities is given in table 1 and are explained below in full. We note that in reality partnership dynamics and trade behaviour depend on a range of factors not considered here, e.g. social networks and capital, but farm-level propensities, supply and demand, capture much of the observed variation in the Scottish cattle trade system (§4).

### 2.1.1. Dynamics of trading partnerships

The evolution of the topology of the modelled system is determined entirely by the formation and cessation of trading partnerships. Under the model, each farm possesses a dynamic list detailing which farms they can purchase animals from at a given time. Purchasing farms continually seek to optimize their trading partners by preferentially forming partnerships with large suppliers, i.e. farms with large $\zeta_i$, and preferentially ending partnerships with small suppliers, such that the system tends towards an equilibrium in which farms maintain long-lasting partnerships with large suppliers. A farm $i$ begins a trading partnership with another farm $j$, given no current partnership between them, at rate

$$\alpha_{ij} = \frac{a_i}{N} \eta_i \zeta_j,  \tag{2.1}$$

where constant $a_i$ represents the propensity for farm $i$ to form trading partnerships, summarizing all factors that impact the ability of farm $i$ to do so, e.g. the time required to search for partners. This process is uni-directional and, in general, asymmetric ($\alpha_{ij} \neq \alpha_{ji}$).

A current trading partnership between farms $i$ and $j$ ends at rate

$$\delta_{ij} = \frac{d_i}{\eta_i \zeta_j}  \tag{2.2}$$

such that all farms tend to maintain longer partnerships with large suppliers compared with smaller suppliers. High-demand farms are less likely to end trading partnerships in general compared to low-demand farms. The constant $d_i$ represents an intrinsic measure of the propensity for farm $i$ to remove one of its traders, with larger values resulting in shorter duration trade partnerships, and vice versa.

The equilibrium probability of there being a trading partnership between $i$ and $j$ is $p_{ij}$, and the expected number of trading partners for farm $i$, $k_i^{in}$, is calculated as shown in table 1 (see electronic supplementary material, §1 for further details). The $1/N$ scaling of $\alpha_{ij}$ in (2.1) ensures that $k_i^{in}$ does not scale linearly with the system size, $N$.

### 2.1.2. Movement of animals and trade flows

Animals are assumed to move between trading partners from $j$ to $i$ in batches (the number of individual animals moved in a single trade) of constant size $\theta_i$ with rate

$$\varphi_{ij} = b_i \min(\eta_i, \zeta_j),  \tag{2.3}$$

where $b_i$ is taken to represent any impediment to the movement of animals, for example delivery of livestock. The second term in (2.3) is referred to as the reference transaction rate and is the maximum

**Table 1.** Table of model quantities and their respective definitions.

| quantity | definition |
|---|---|
| $N$ | number of farms |
| $\eta_i$ | annual in-flow of animals for farm $i$ |
| $\zeta_i$ | annual out-flow of animals for farm $i$ |
| $a_i$ | rate describing farm $i$'s propensity to form trading partnerships |
| $d_i$ | rate describing farm $i$'s propensity to end trading partnerships |
| $\alpha_{ij} = a_i \eta_i \zeta_j / N$ | rate at which $i$ forms a trading partnership with $j$ |
| $\delta_{ij} = d_i / (\eta_i \zeta_j)$ | rate at which $i$ ends a trading partnership with $j$ |
| $p_{ij} = \alpha_{ij} / (\alpha_{ij} + \delta_{ij})$ | the probability that a trading partnership made by $i$ with $j$ is present |
| $k_i^{in} = \sum_{j \neq i}^{N} p_{ij}$ | expected instantaneous number of concurrent trading partners for farm $i$ conditioned on zero partnerships at $t = 0$ |
| $b_i$ | rate describing farm $i$'s propensity to initiate trades with its trading partners |
| $\varphi_{ij} = b_i \min (\eta_i, \zeta_j)$ | rate at which $i$ trades with its trading partner $j$ |
| $\theta_i$ | batch size for farm $i$ |
| $V_i^{in} = \theta_i \sum_{j \neq i}^{N} \varphi_{ij} p_{ij}$ | expected unit-time equilibrium in-flow of animals for farm $i$ |
| $\lambda$ | disease prevalence on an infected farm |
| $B(\theta_i) = 1 - (1 - \lambda)^{\theta_i}$ | probability at least one infected animal moves on to a susceptible farm $i$ given batch size $\theta_i$ |
| $\beta_{ij} = \varphi_{ji} B(\theta_j)$ | transmission rate from infected farm $i$ to susceptible farm $j$, given a trade partnership currently exists between farms $i$ and $j$ |
| $\gamma$ | disease recovery rate |

rate of exchange of indivisible goods (livestock), since $1/\eta_i$ is the expected time for $i$ to generate new demand for animals and $1/\zeta_j$ the expected time for $j$ to generate new supply [37,38].

The per unit time in-flow of animals for farm $i$, when the system is at equilibrium, which is expected to equal $\eta_i$, is

$$V_i^{in} = \eta_i = \theta_i \sum_{j \neq i}^{N} \varphi_{ij} p_{ij}. \tag{2.4}$$

This expression is easily interpreted, since $\varphi_{ij} p_{ij}$ is the expected number of trades from $j$ to $i$ in a unit of time, and $\theta_i \varphi_{ij} p_{ij}$ is the total number of animals $i$ purchased from $j$. Summed over the entire system, we obtain the total in-flow of animals per unit time for farm $i$. This expression for $V_i^{in}$ allows us to alter the dynamics of trading partnerships and the movement of animals while maintaining each farm's in-flow of animals. We shall explore the effect of such conservative changes in §3.

### 2.1.3. Disease dynamics

The dynamics of disease are coupled with partnership dynamics and trade behaviour by assuming disease is driven entirely by animal movements, neglecting indirect transmission such as from external wildlife sources or distance modulated local infection.

We categorize disease status at farm level using a standard susceptible–infected–susceptible (SIS) model; susceptible farms become infected through trade with infected farms, can themselves infect others and, after an exponentially distributed infectious period with mean $1/\gamma$, recover to become susceptible once again. In addition to the infectious period, a given disease is also characterized by an effective on-farm prevalence level $\lambda$, assumed constant across infected farms and time. We therefore take $\lambda$ to be the average prevalence of an infected farm over its infectious lifetime. We assume each animal moved off an infected farm $i$ has a constant probability $\lambda$ of infecting the susceptible buying farm and that off-farm movements do not alter herd prevalence on the selling farm. If an infected farm sells $\theta$ animals in a trade to a susceptible farm, the total probability of transmission is $B(\theta) = 1 - (1 - \lambda)^{\theta}$, and the rate at which a farm $j$ receives infection from its infectious trade partner $i$

is $\beta_{ij} = \varphi_{ji}B(\theta_j)$, i.e. the rate at which $j$ trades with $i$ multiplied by the probability that the trade results in the transmission of disease. Thus, trades that occur with large size are more likely to result in the transmission of disease.

# 3. Results

## 3.1. Farms' basic reproduction number

Calculating $R_0$ for our model is challenging due to the heterogeneous nature of partnerhip dynamics and trading. Furthermore, the central role of the partnership network in mediating trade invalidates possible assumptions of homogeneous mixing. However, the methods outlined in [39] allow for an expression for $R_0$ to be obtained by considering the dynamics of farm pairs and calculating the probability of disease transmission. We extend these methods by incorporating farm heterogeneities and deriving a per-farm expression for $R_0$, $R_0^i$. Details of the calculation are provided in electronic supplementary material, §2, but assume that the trading sub-system has reached an equilibrium (true for all simulations presented) and the partnership network is sufficiently sparse. The latter condition is satisfied since, for large systems, the probability of a two-way trading partnership scales as $1/N^2$. It is important to note that the results presented do not depend on the functional forms adopted above to describe partnership dynamics and trade behaviour and so offer general insights.

For a large system, $R_0^i$ reduces to

$$\lim_{N \to \infty} R_0^i = \sum_{j \neq i}^{\infty} p_{ji} T_{ij} + \sum_{j \neq i}^{\infty} \frac{\alpha_{ji}}{\gamma} T_{ij}, \tag{3.1}$$

(see electronic supplementary material, §2), where the transmissibility

$$T_{ij} = \frac{\beta_{ij}}{\beta_{ij} + \delta_{ji} + \gamma}$$

is the probability that farm $i$ infects farm $j$ if there is a trading partnership present, before the end of the infectious contact period, i.e. prior to either recovery or the ending of the partnership [40]. The first term in (3.1) accounts for the number of current trade partnerships that result in the transmission of disease. The second term accounts for the number of new trade partnerships formed, before $i$ recovers, that result in disease transmission before the end of the infectious contact period. This shows that partnership dynamics play a significant role in the ability for an infected farm to make infectious contacts. Indeed, even if the transmissibility was set to unity, so that farm $i$ was guaranteed to pass infection on to its buyers following a trade, $R_0^i$ would still be bounded by the rate at which buying farms sought out new trade partnerships with $i$, i.e. by $\alpha_{ji}$.

## 3.2. The effect of changes to trading practices

We now use the above expression of $R_0^i$ to rigorously explore the effects of modifying trading practices under the strong constraint (2.4) that farms maintain their expected in-flow of animals. Illustration of these results using stochastic simulations of example systems are presented in electronic supplementary material, §3.

### 3.2.1. The role of trade behaviour

Consider first changes to the frequency and size of trades. Due to (2.4), and supposing the dynamics of trade partnerships are kept constant, a linear increase in the frequency of trade is accompanied by a proportional decrease in the size of trades, and vice versa. We introduce the scaling parameter $\varepsilon_{\text{trade}}$ that determines the frequency and size of trades, and set

$$\varphi_{ij} \to \varepsilon_{\text{trade}} \varphi_{ij}$$

and

$$\theta_i \to \varepsilon_{\text{trade}}^{-1} \theta_i$$

for all $i$ and $j$. Considering the case of large trades, substitution into the transmissibility, $T_{ij}$, reveals

$$\lim_{\varepsilon_{\text{trade}} \to 0} T_{ij} = \lim_{\varepsilon_{\text{trade}} \to 0} \left( \frac{\varepsilon_{\text{trade}} \varphi_{ji} B(\varepsilon_{\text{trade}}^{-1} \theta_j)}{\varepsilon_{\text{trade}} \varphi_{ji} B(\varepsilon_{\text{trade}}^{-1} \theta_j) + \delta_{ji} + \gamma} \right) = 0,$$

since $B(\varepsilon_{\text{trade}}^{-1} \theta_j)$ is bounded above by 1. It immediately follows that

$$\lim_{\varepsilon_{\text{trade}} \to 0} \lim_{N \to \infty} R_0^i = 0 \tag{3.2}$$

for all $i$. Thus, increasing the batch size reduces $R_0$. Similarly, in the case $\varepsilon_{\text{trade}} \to \infty$ where trades occur more frequently, but take ever smaller size, we find that $R_0^i$ approaches a well-defined non-zero limit, further confirming that disease spread is inhibited by the dynamics of trade partnerships. This is due to the conservation of the in-flow of animals, so that the infection rate $\beta_{ij}$ does not scale linearly with $\varepsilon_{\text{trade}}$, but rather approaches a limit given by $\varphi_{ji} \theta_j \ln(1/(1-\lambda))$, implying that although the number of trades increases significantly, the force of infection does not rise indefinitely due to the decrease in batch size. See electronic supplementary material, §2.1 for details.

### 3.2.2. The role of partnership dynamics

We now explore the dynamics of trade partnerships when the frequency and size of trade is fixed. To do so, we introduce the scaling constant $\varepsilon_{\text{ptnr}}$ and set

$$\alpha_{ij} \to \varepsilon_{\text{ptnr}} \alpha_{ij}$$

and

$$\delta_{ij} \to \varepsilon_{\text{ptnr}} \delta_{ij},$$

which allows for the dynamics of trade partnerships to be explored while maintaining a farm's expected instantaneous number of trading partners, $k_i^{in}$. As $\varepsilon_{\text{ptnr}}$ increases, partnerships are formed increasingly frequently; however, these partnerships last a decreasing period of time, and vice versa. In these limits, we obtain

$$\lim_{\varepsilon_{\text{ptnr}} \to 0} \lim_{N \to \infty} R_0^i = \sum_{j \neq i}^{\infty} \frac{\alpha_{ji}}{\delta_{ji}} \frac{\beta_{ij}}{\beta_{ij} + \gamma} \tag{3.3}$$

for long-duration partnerships, which is equivalent to the value of $R_0$ for a static directed network [32], so that the spread of disease is entirely dependent on the initial distribution of trade partnerships mediated by trade between them. We note that this is the scenario explored by [38]. Similarly, for small-duration partnerships, we obtain

$$\lim_{\varepsilon_{\text{ptnr}} \to \infty} \lim_{N \to \infty} R_0^i = \sum_{j \neq i}^{\infty} \frac{\widehat{\beta_{ij}}}{\gamma}, \tag{3.4}$$

where $\widehat{\beta_{ij}} = \beta_{ij} \alpha_{ji} / \delta_{ji}$, which is equivalent to the value of $R_0$ for a system under the mean-field assumption. Comparing (3.3) and (3.4), since $\beta_{ij} + \gamma > \gamma$ for all $\beta_{ij} > 0$, the disease is expected to spread more prolifically when trade partnerships are temporary, and a static network approximation offers a lower bound on the early-time spread of disease, if all other components of the system are kept constant.

### 3.2.3. The role of the number of concurrent trading partners

Finally, we consider the effect on $R_0^i$ of changes to the number of concurrent trading partners. Since there are an infinite number of combinations of $\alpha_{ij}$ and $\delta_{ij}$ that result in a given $k_i^{in}$, here we fix the duration of trade partnerships, i.e. keep $\delta_{ij}$ constant, and set

$$\alpha_{ij} \to \varepsilon_{\#\text{ptnr}}^{ij} \alpha_{ij}.$$

Note the $i, j$ dependence of $\varepsilon_{\#\text{ptnr}}^{ij}$ in this case. We also note that conservation equation (2.4) implies a change in the number of trading partners must be accompanied by an inverse change in either the trade rate $\varphi_{ij}$ or the batch size $\theta_i$ (or both). For simplicity, we herein maintain (2.4) by fixing the batch size and increasing/decreasing the trade rate when the number of trading partners is altered.

For a proportional change in $k_i^{in}$ of $x$, we have

$$\varepsilon_{\#ptnr}^{ij} = \frac{x\alpha_{ij}}{(1-x)\alpha_{ij} + \delta_{ij}},$$

which can be verified by substitution into our expression for $k_i^{in}$ (table 1). In the limit of a small number of concurrent trading partners, we find

$$\lim_{\varepsilon_{\#ptnr}^{ji} \to 0} \lim_{N \to \infty} R_0^i = 0 \tag{3.5}$$

as expected since the system becomes entirely disconnected. For the scenario in which the number of concurrent trading partners goes to $N$, as $N$ increases so too does $R_0^i$. As such, we use the expression for $R_0^i$ for a system of finite size (see electronic supplementary material, §2.4), and obtain

$$\lim_{\varepsilon_{\#ptnr}^{ji} \to \infty} R_0^i = \sum_{j \neq i}^{N} \frac{\beta_{ij}}{\beta_{ij} + \gamma}. \tag{3.6}$$

Note here that even for a finite system to reach this limit, $\varepsilon_{\#ptnr}^{ij}$ must go to infinity as the partnership cessation rate is fixed. Unsurprisingly, when the system is completely connected, the spread of disease is dependent solely on the dynamics of trade and the intrinsic disease parameters.

# 4. Case study: Scottish cattle trade industry

We demonstrate the potential of our modelling framework by application to the Scottish cattle trade system. We first show it is able to capture key features of this complex real-world system, and then use it to assess the potential impact of changes to trade patterns for the Scottish cattle industry. We use data from the Cattle Tracing System (CTS) for 2005–2013 inclusive, avoiding perturbations resulting from restocking following the UK 2001 foot-and-mouth Disease (FMD) outbreak [29]. We focus on the Scottish subset of this dataset featuring 15 386 cattle farms which engage in a total of 135 106 trades per year, with a total of 420 931 animal moves per year averaged over 2005–2013. We consider this a closed system, ignoring in-flow (representing approx. 10% of on-movements) and out-flow (approx. 14% of off-movements) of animals beyond Scotland, and consider only farm-to-farm movements grouped into dated batches. Animal flows through markets are maintained by treating such movements as transitory and replacing them with direct farm-to-farm movements. Movements to market are expected to play a small role in direct transmission of endemic livestock disease [31,41], but we acknowledge for epidemic spread of exotic or re-emerging diseases, market transmission may play a more significant role, for example in the 2001 FMD epidemic [11]. As such, we consider only slow spreading endemic diseases.

The farm-to-farm batch movement data described above are used to parametrize our model as follows (further details and distributions of trade quantities are presented in electronic supplementary material, §4). Electronic supplementary material, figure S5 shows trading patterns and animal flows are consistent year-on-year (movements at farm level are also known to be consistent year-on-year [31]), and we obtain annualized average in- and out-flows, $\eta_i$ and $\zeta_i$, for each farm by averaging observed yearly numbers of animals purchased and sold, respectively. As above, the batch size for farm $i$, $\theta_i$, is assumed constant, independent of the originating farm, and is estimated from data by averaging the total in-flow over the total number of trades for each farm.

Estimates for the trade partnership formation and cessation constants $a_i$ and $d_i$ are determined by evaluating partnerships on an annual basis, that is for a given year a partnership exists where two farms trade in that year. From the data, we find that 83% of trade partnerships end after a single year, and 89% end after 2 years, emphasizing the importance of accounting for partnership dynamics. To calculate $a_i$, we match observed new trading partners from year $t$ to year $t+1$ with the partnership formation rate defined in (2.1), averaged over all years. Similarly, the constant $d_i$ in the partnership cessation rate (2.2) is found by equating the number of partnership cessations occurring from one year to the next. Finally, the constant $b_i$ in the trade rate (2.3) is obtained by solving the constraint equation (2.4) given estimates for all other quantities. Distributions across farms for each of these quantities can be found in electronic supplementary material, §4.1.

Initial results based on the above parameter estimates obtained for the model described in §2 reveal that our proposed trading partnership formation and cessation rates did not accurately replicate the distributions of the duration of trade partnerships or the joint distribution of farms' in-flows, $\eta_i$, and

their traders' out-flows, $\zeta_j$. We therefore modified these rates to

$$\alpha_{ij} = \frac{a_i}{N}(\eta_i \zeta_j^m + w) \tag{4.1}$$

and

$$\delta_{ij} = \frac{d_i}{\eta_i}, \tag{4.2}$$

and find that setting $m = 0.75$ and $w = 75$ yields results closer to those observed in the data as shown in figure 1 (initial fits are presented in electronic supplementary material, figures S10 and S11), while also replicating the values of higher-order statistics, e.g. annual in-flow, number of concurrent trading partners, and number of trades. This indicates the flexibility of our approach to represent real-world complexity in a parsimonious and tractable generative model framework. The required modifications to the model rates show that small buyers place greater weight on factors other than simply the size, $\zeta_i$, of the prospective seller, but that larger buyers tend to buy from larger suppliers. Furthermore, large sellers are, in general, kept as trading partners for the same period of time as small sellers, again suggesting that farm sizes (the volume of animals bought/sold) are only one factor in selecting trade partners.

## 4.1. Assessing the potential for trade practices to modulate endemic disease

We now explore the effect of increased trade size, longer duration of trade partnerships, and reduced number of concurrent trading partners, subject to the constraint that farms' in-flows are maintained. To do so, we focus on a fixed disease parametrization $\lambda = 0.2$ and $1/\gamma = 5$ years, which is intended to represent a high-prevalence, high-persistence disease. For this hypothetical disease parametrization and current Scottish trading patterns, our model predicts a system-average $R_0^i$ $R_0 \approx 10$.

Figure 2 shows the percentage reduction in $R_0$ under varying changes to trade and trade partnership dynamics compared with current trading patterns (see electronic supplementary material, figure S14 for $R_0$ values). This shows that fewer, longer lasting trade partnerships yield the greatest reduction in $R_0$, with up to 90% reduction when farms maintain a single, near-permanent trade partner. Fewer concurrent partnerships combined with fewer, larger trades reduce $R_0$ by up to 76%; however, reducing the number of concurrent partnerships is responsible for most of this reduction. In the Scottish trading system, cattle farms average approximately 7.3 concurrent annual trading partners, and batches take average size of 3.58. Changes to current partnership dynamics and trading behaviour could yield both significant reductions and increases in $R_0$. For example, if the system-average number of concurrent trading partners and batch size were reduced by one, then $R_0$ would be reduced by approximately 12%. Conversely, if these were to be increased by one, then $R_0$ is increased by over 15%.

## 4.2. Impact of trade practices on a wide range of endemic diseases

We now explore the effect of specific changes to trade and partnership dynamics for a broad range of disease parametrizations (see electronic supplementary material, figure S15 for $R_0$ values). We consider halving the average number of concurrent trading partners, doubling the duration of trade partnerships, and doubling the average batch size, with each of these interventions considered under every possible combination (figure 3) and in isolation (electronic supplementary material, figure S16). These changes are again made subject to conserving individual farms' in-flows of stock. Chosen farm-level prevalence, $\lambda$, ranges from 0.01 to 0.25, with infectious periods, $1/\gamma$, ranging from six months to 5 years.

Changes to the size and frequency of trades are most effective in reducing $R_0$ for high-prevalence, small-duration diseases, whereas changes to the duration and number of trade partnerships are most effective on high-prevalence, long-duration diseases (see electronic supplementary material, figure S16). This difference is explained by the fact that as the batch size increases, the inter-trade times increase, so that for small-duration diseases the probability that an infected farm recovers before it is traded with increases. Changes to multiple aspects of trade patterns yield greater reductions in $R_0$ compared with changes to single elements. Encouraging fewer, longer lasting trade partnerships combined with fewer, larger trades provides the greatest reduction in $R_0$ (up to 53% for the highest prevalence and longest lasting diseases considered here) and also bring $R_0$ below 1 for a greater range of diseases. It is noteworthy that our suggested changes bring $R_0$ below 1 for diseases that are already close to this threshold, but also significantly reduce $R_0$ for high-prevalence, long-duration diseases, i.e. diseases that are extremely challenging to control and eradicate.

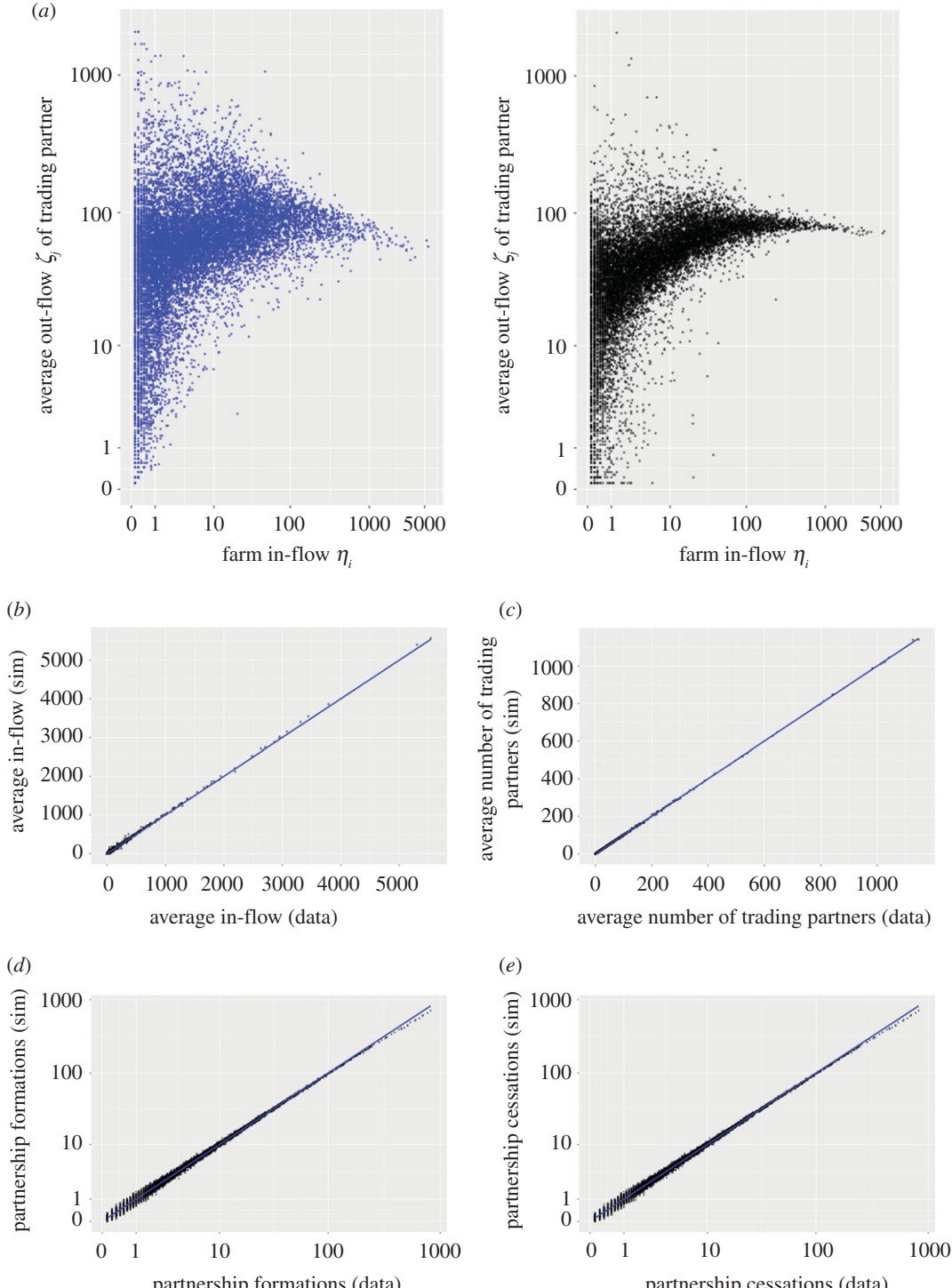

**Figure 1.** Model fit to data. For model with modifications to partnership formation and cessation rates. Panel (*a*) shows the average out-flow, $\zeta_j$, of farms' trading partners, where blue points are obtained from data, and black points from stochastic simulation, where simulations are performed using Gillespie stochastic simulation algorithm. Bottom four panels show comparisons of simulation output and data for four statistics: annual in-flow (*b*), annual number of concurrent trading partners (*c*), annual number of partnership formations (*d*) and annual number of partnership cessations (*e*).

## 4.3. Targeting the trade practices of large buyers

The results above show significant reductions in $R_0$ are attainable when all farms change their trade behaviour and partnership dynamics. However, consistent with other livestock markets [29,38], the Scottish trading system exhibits scale-free-like properties; a small number of farms trade much more frequently than the average and have a much larger annual number of concurrent trade partners (see

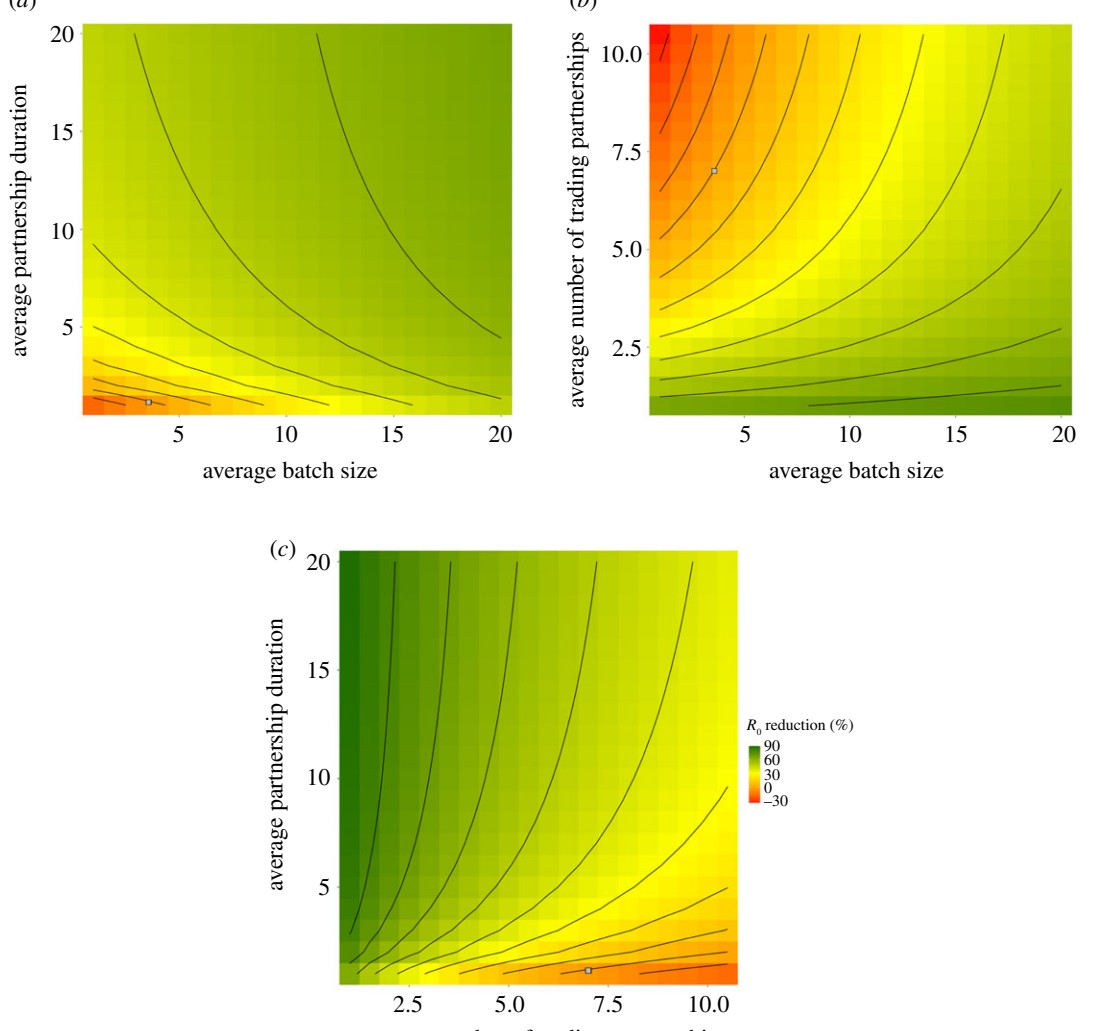

**Figure 2.** Impact of trade behaviour and partnership dynamics. Percentage change in $R_0$ for a persistent and high-prevalence disease ($\lambda = \gamma = 0.2$) due to changes in the dynamics of trade and trade partnerships compared with the current dynamics of the Scottish trade system (grey squares). We consider changes to batch size and partnership duration (a), batch size and number of concurrent trading partners (b), and number of concurrent trading partners and partnership duration (c).

electronic supplementary material, figures S7 and S8). Despite this, these outlying farms have average batch sizes similar to the mean batch size (and in some cases smaller, for example the 1% of farms that make the largest number of trades make, on average, 363.5 trades per year, with average batch size 2.86, whereas the mean batch size is 3.58), suggesting there is scope for such farms to increase their average batch size. We therefore explore the potential for changes targeted at the most frequent buyers (those farms making the largest number of trades annually) and compare the resulting system average $R_0$ with the value of $R_0$ for current (i.e. no changes to) trade patterns, and with the value of $R_0$ obtained when all farms adopt the proposed changes.

Figure 4 shows the results from targeting the top $x\%$ of farms with $x$ ranging from 0 to 100%. The changes to trading patterns considered are the composite changes that lead to the greatest reduction in $R_0$ in figure 3. These changes are assessed under three disease parametrizations: case (1) $\lambda = 0.06$, $\gamma = 1$, corresponding to a disease scenario in which our suggested changes in §4.2 brought $R_0$ below 1, case (2) $\lambda = 0.15$, $\gamma = 0.4$ and case (3) $\lambda = 0.25$, $\gamma = 0.2$, corresponding to the disease parametrization that provided the greatest reduction in $R_0$ for the range of parameters we explored in §4.2.

In all disease scenarios, 20% of the most frequent buyers are responsible for approximately 87% of the total possible reduction in $R_0$. Moreover, when 50% of the most frequent buyers adopt the proposed changes to trading patterns, we obtain approximately 98% of the reduction in $R_0$ that would be achievable if all farms comply. In case (1), 8% compliance is sufficient to bring $R_0$ below 1, suggesting that for diseases with values of $R_0$ close to the threshold value, only a small fraction of farms would

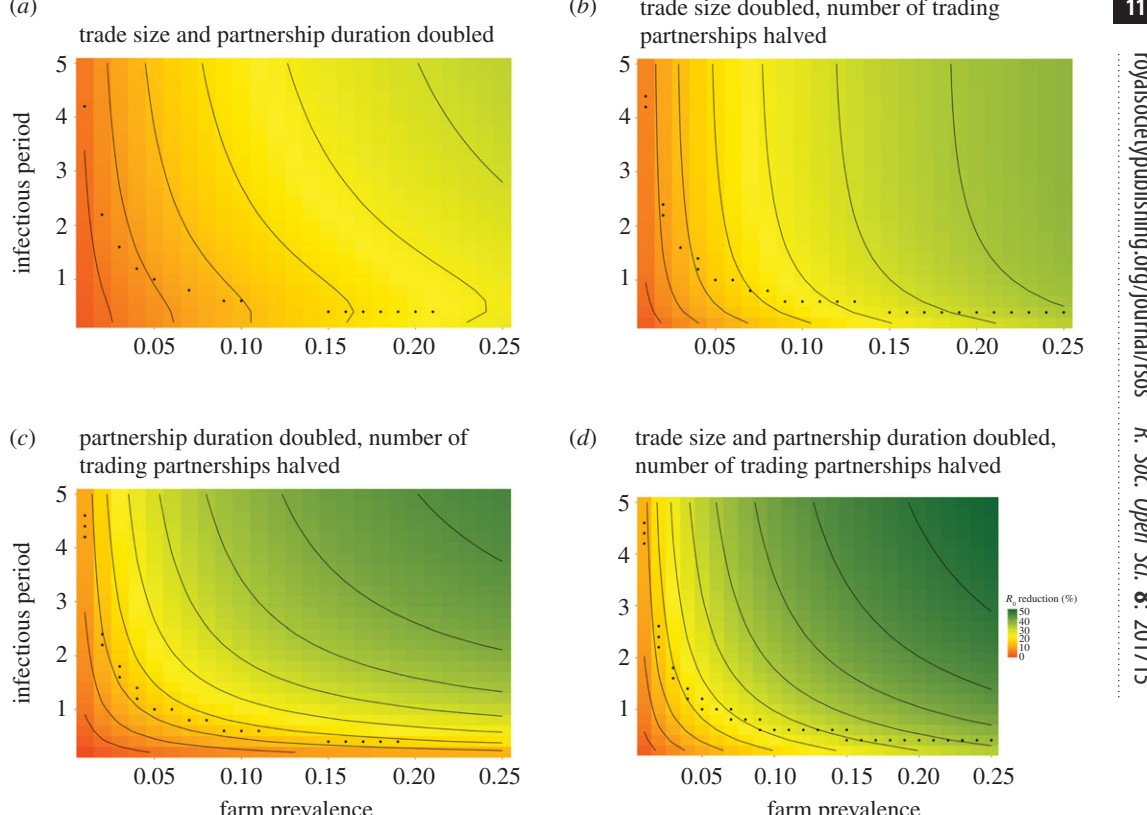

**Figure 3.** Reducing endemic disease burden. The percentage reduction in the system average $R_0$ for a range of disease parametrizations under specific trading and partnership dynamics changes, when compared with values of $R_0$ for current trading patterns in the Scottish trade system. Black points represent disease parametrizations in which $R_0 > 1$ before changes, and $R_0 < 1$ after changes are implemented.

need to change their trading patterns to eradicate disease. For diseases that are challenging to control (case (3)), significant reductions are still achievable through the targeted approach, though stricter control measures may be necessary to bring $R_0$ below 1 for these diseases.

## 4.4. Combining targeted changes to trading practices with targeted biosecurity

So far we have considered only changes to buyers' trading patterns, but now show that targeted changes in trade may be more impactful than similar targeting of standard on-farm biosecurity measures. We assess the impact of varying percentages of the largest sellers (those with the largest annual out-flow of animals) adopting on-farm biosecurity that is assumed to reduce prevalence $\lambda$ and the infectious period $1/\gamma$ from a baseline ($\lambda = 0.25$ and $1/\gamma = 5$). These targeted biosecurity changes are assessed alone and in combination with changes to trading patterns targeted at the most frequent buyers, as above. Figure 5 shows that the combination further reduces system average $R_0$ compared to solely targeting trade patterns. However, these additional reductions increase relatively linearly as an increasing fraction of sellers adopt improved biosecurity. This is in stark contrast to the impact of an increasing fraction of the largest buyers changing trade practices (figure 4) for which most of the potential reduction in $R_0$ is due to a small fraction of the most frequent buyers. This may be understood by considering that our analysis of the Scottish trading system suggests that formation and cessation of trading partnerships is determined by more factors than simply the size of the selling farm, i.e. their $\zeta_i$. Thus, the out-flow of animals of a farm does not solely indicate whether that farm is a potential risk for the spread of disease.

## 5. Discussion

Animal movements via trade have long been considered a significant factor in the spread and persistence of diseases within national scale livestock disease systems [1–8]. Recently, available movement data have enabled modelling of disease spread to be superimposed on historic livestock movement patterns

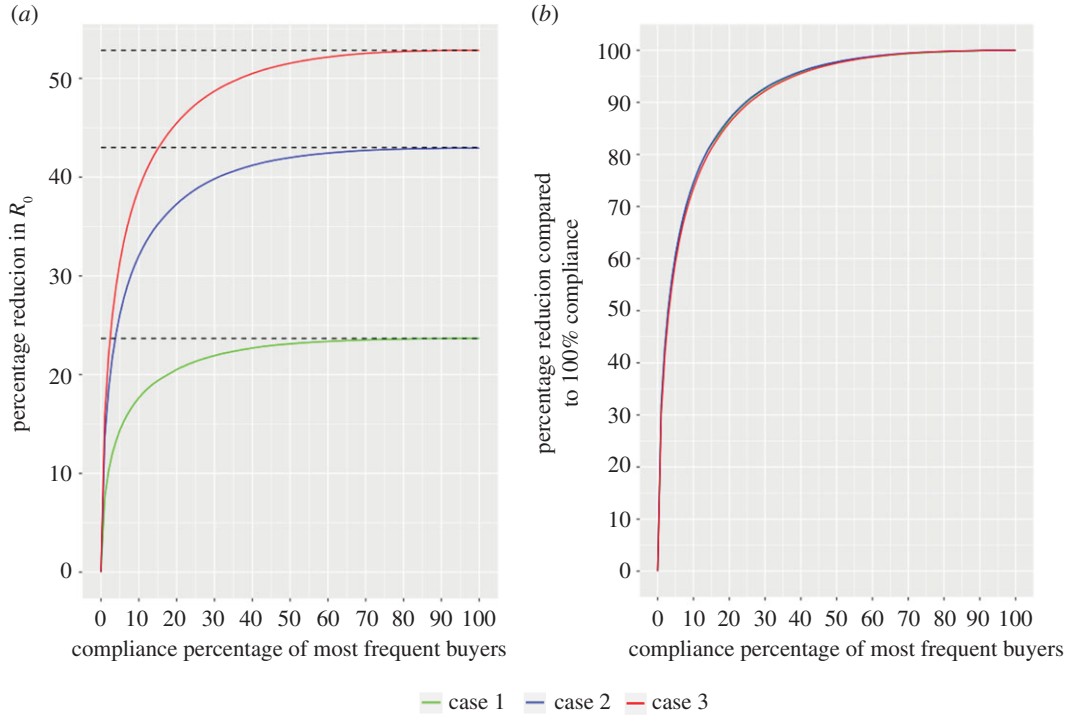

**Figure 4.** Targeting high-risk farms. Percentage reduction in $R_0$ compared to: current trading patterns (*a*); and 100% adoption of new trading patterns (*b*). The new trading patterns are those shown in figure 3*d*, which provides percentage reduction at 100% (dashed lines). In both panels, the *x*-axis indicates what percentage of the most frequent buyers (those making the largest number of trades annually) are adopting these changes. Different disease parametrizations are shown with dashed lines representing values of $R_0$ for: case (1) $\lambda = 0.06$, $\gamma = 1$; case (2) $\lambda = 0.15$, $\gamma = 0.4$; and case (3) $\lambda = 0.25$, $\gamma = 0.2$. Initial $R_0$ values for current trading patterns are: case (1) $R_0 = 1.19$, case (2) $R_0 = 4.92$ and case (3) $R_0 = 11.43$.

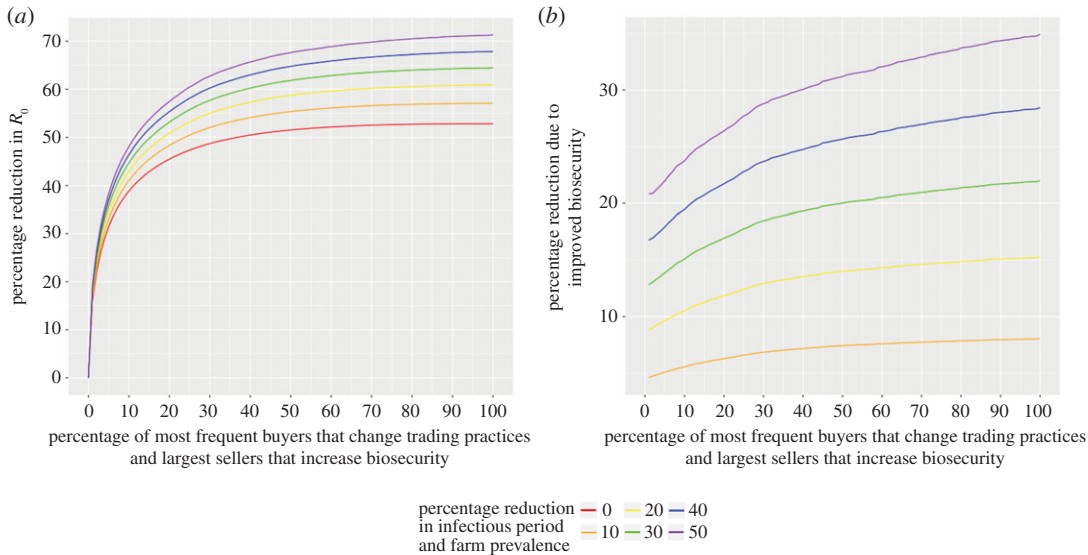

**Figure 5.** Comparing biosecurity with changes to trade patterns. Percentage reduction in $R_0$ (disease parameters $\lambda = 0.25$ and $\gamma = 0.2$) for targeted changes to both trading practices and improved biosecurity (*a*), and due solely to targeted improvements to biosecurity (*b*). In both cases, *x*-axes indicate what percentage of the most frequent buyers adopt trade changes and largest sellers improve biosecurity.

[4,42,43]. Network analysis of such data have also proved highly insightful. For example, using static networks to identify that fewer larger trades could improve disease control [44], or that highly connected 'hubs' are likely efficient targets for biocontrol [31]. Nonetheless, there is a pressing need to develop truly generative models of livestock movements to enable such data to better inform

understanding and management of these complex systems. In this article, we outline a generative approach with two components: a dynamic network which evolves via continuous formation and cessation of trading partnerships determining network topology at a given time; and a contact process on this network that represents animal movement (trades) and related disease spread between farms. Our approach goes beyond current state-of-the-art models [38], for which only the size and timing of animal movements is modelled on a fixed network of trade partnerships, and is sufficiently powerful to represent key features of Scottish cattle movements as recorded by the Cattle Tracing System (CTS). Analysis of this model yields powerful insights into disease control, with limiting cases allowing re-derivation of known $R_0$ expressions, e.g. for static networks and well-mixed systems.

In the context of the Scottish cattle trading system, we show that disease risks can be reduced in a way that minimizes disruption by maintaining annual in-flows of animals for all farms. Fewer, larger trades and fewer, longer lasting trade partnerships yield the greatest reduction in system average $R_0$ when they are applied simultaneously, especially for diseases with high prevalence and persistence. Moreover, they can reduce $R_0$ below 1 for diseases close to this critical threshold under current trading patterns. Thus, changes in trade practices could eradicate certain diseases without other, potentially more disruptive and costly, control measures, and they could assist control of more persistent diseases that require multiple interventions. The fact that $R_0$ can be significantly reduced by simply changing the ways in which farms maintain their annual in-flow of animals is, we believe, a significant finding as this is potentially far less intrusive than other control strategies involving, for example, movement bans or restricting from whom a farm can purchase animals [35]. We note, however, that different network structures may effect the efficacy of each of our proposed changes to trade.

Our analysis also highlights the potential to exploit scale-free-like properties of livestock trading systems for disease control. Targeted changes to the trade practices of only the farms with the highest trade volumes can significantly reduce $R_0$ and thus the burden of endemic disease and outbreak risk for the whole system. Further reductions result from combining changes to trade patterns with more standard biosecurity measures targeted on farms with the largest annual out-flows of animals. As such targeted modifications are expanded, resulting disease control benefits from changing trade practices scale much more favourably than do those of similarly targeted farm-level biosecurity (figure 5). Given the current emphasis on farm-level biosecurity this is further evidence that the disease control potential of modifying trade deserves greater attention.

These results illustrate how mechanistic generative models, such as introduced here, can make a unique contribution to the study of livestock networks that complements existing network approaches. For example, our results agree with static network analysis identifying that fewer larger trades could improve disease control [37,44], but go beyond these to show the impact of trade partnership dynamics. The scale-free properties of livestock trade are a common target for network analysis including recent work on UK livestock trade that shows a fraction of farms are highly connected by contact chains involving multiple trades [31]. Although we do not explicitly identify such contact chains, their influence on disease transmission is integrated into our analysis and captured in our calculations of $R_0$ that account for trade and the formation of trade partnerships.

Naturally, the first implementation of our novel framework has made simplifying assumptions, the relaxation of which will be the subject of further work. Firstly, we assume trade occurs throughout the year; however, animal movements generally occur in specific months [29]. Secondly, the rate at which farms trade is assumed constant, regardless of when the last trade was, but fluctuation in supply and demand is likely to play an important role in trade dynamics. However, we note that currently available generative mechanistic models of livestock trade make similar assumptions [37,38]. Reformulating the trade rate to be a function of these stock quantities is a natural progression of our model which would resolve these issues, but could limit analytic tractability. Finally, the rates determining the formation and cessation of trade partnerships are based only on the annual in- and out-flows of farms, but our analysis suggests other factors may be at play. Distance-based metrics, farm types (beef, dairy, etc.), time-varying stock rates (see above) and socio-economic factors may enable better quantification of trading and partnership dynamics, and may also prove significant in the spread of disease.

In conclusion, we have introduced what we believe is the first generative modelling framework for livestock movements that is able to account for key features of complex national scale real-world systems. Analysis of resulting between-farm disease spread shows changes to trading patterns that conserve farm-level in-flow of animals provide a powerful approach to control of endemic disease and probably also mitigate outbreak risk. Attempts to adopt these novel approaches to disease control may reveal frictions in the ability of a real-world trading system to implement our proposed changes to trade and further work is needed to explore such barriers to uptake. For example, larger batch sizes

(and fewer trades) may inhibit flexibility in adapting to changing conditions. Furthermore, there is evidence that some farmer behaviours are determined by responses to external influences including extreme weather events and socially accepted farming practices [45]. This suggests that incentives, e.g. in the form of cooperatives, health schemes, or subsidies, may be required to encourage modification of farm-level trading behaviour. However, it is encouraging that reductions in disease burden resulting from targeted modification of trade practices scale much more favourably than those associated with improvements to farm biosecurity that are the usual focus of disease control policies.

Data accessibility. All code required to run the model is freely available at https://github.com/MKnight-bioss/generative_paper. Cattle Tracing System data contains all movements of animals In Great Britain, as well as confidential information of individuals, e.g. addresses, IDs. The data are managed by Defra, and with the agreement of the Editorial Office, the authors cannot make the data publicly available. Individuals wishing to request access to the data can contact Andy Mitchell (andy.mitchell@apha.gsi.gov.uk).

Authors' contributions. The work was planned and the manuscript was prepared by M.A.K., P.C.L.W., R.S.D., M.R.H and G.M. Initial models were created and exploratory analysis was conducted by G.M. and R.S.D. All code was written by and simulations and analysis were performed by M.A.K. All authors gave final approval for publication.

Competing interests. We declare we have no competing interests.

Funding. G.M., R.S.D. and M.R.H. are supported by the Scottish Government's Rural and Environment Science and Analytical Services Division (RESAS). M.A.K. was supported by a studentship funded by Biomathematics and Statistics Scotland, SRUC, University of York and Mains of Loirston Charitable Trust (Scottish Registered Charity, Number SC038006).

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
