## [Peer Review File · Royal Society Open Science]

Review History

RSOS-201715.R0 (Original submission)

Review form: Reviewer 1

Is the manuscript scientifically sound in its present form?

Yes

Are the interpretations and conclusions justified by the results?

Yes

Is the language acceptable?

Yes

Do you have any ethical concerns with this paper?

No

Have you any concerns about statistical analyses in this paper?

No

Recommendation?

Accept with minor revision (please list in comments)

Comments to the Author(s)

I enjoyed reading this paper, finding it well written and easy to follow. I only have 2 comments that are not all that substantive:

1) In figure 2, the contrast showing the reduction in R_0 is difficult to see. I would suggest using a different colour map rather than shades of a single colour.

2) I have one comment on the units of some of the variables used in this paper: The rate at which farms form/dissolve partnerships is given in terms of inflows and outflows of animals. The units of these rates must be the number of partnerships formed/dissolved per unit time but the formation rate is proportional to η times ζ while the dissolution rate is inversely proportional to this product so I cannot see how the units are consistent unless η , ζ are dimensionless, meaning the coefficients a , d have units of 'per time'. Please can you explain.

Review form: Reviewer 2**Is the manuscript scientifically sound in its present form?**

Yes

Are the interpretations and conclusions justified by the results?

Yes

Is the language acceptable?

Yes

Do you have any ethical concerns with this paper?

No

Have you any concerns about statistical analyses in this paper?

No

Recommendation?

Accept with minor revision (please list in comments)

Comments to the Author(s)

Summary of Paper

The paper introduces a mechanistic model which considers partnership dynamics and trading behaviour within livestock to evaluate their effect on mitigating endemic disease. As the previous state of the art paper only considered the size and frequency of trades on a static network, this model is a new contribution to the literature. R_0 was quantified on a system and individual farm basis, affected by changes in partnership and trading dynamics. The effectiveness of specific partnership and trade changes can be affected by network topology. Using a case study of the Cattle Trading System data in Scotland: fewer, long-lasting trade partnerships and small amounts of large trades can reduce R_0 without reducing the demand of cattle on the farm-scale. These results are encouraging, as these measures may have more of an effect on disease reduction than biosecurity increase and be less intrusive. Incentivising high-risk farms to adopt such changes introduces most of the effect of changing trade and partnerships on the entire network.

Recommendation/Overall Thoughts of Paper

Overall, I recommend this paper to be accepted after a few minor revisions. The intro sets an appropriate background for the reader, summarises recent research, and the novel aspect of the proposed model well. The methodology and derivations are largely sound, with sensible inferences being made from the results in a clearly presented manner. Further work and uncertainties of applying such measures to reduce R_0 in practice are well addressed. My revisions in the comments section are primarily for clarity rather than methodological issues.

However, I have one methodological aspect to note regarding the assumption of treating shipments via markets as transitory. The sources which justify this assumption on line 314 only justify not having markets as nodes for slow-spreading endemic diseases. For faster spreading endemic diseases like foot and mouth disease (FMD) for various countries, transmission can occur within a short period and can influence overall prevalence. I would either cite sources which provide evidence of markets contributing a small role in transmission for fast spread endemic diseases, or clarify that the range of endemic diseases modelled can only be slow-spreading within the text.

Minor Comments

Abstract - I would recommend moving “for a broad range of endemic diseases” (line 22) to the end of the next sentence. This is because “We show how these novel modelling tools enable systems approaches to disease control, using R_0 to explore impacts of changes in trading practices on between-farm prevalence levels” refers to the work in section 3, which was only done for one value for the on-farm prevalence and removal rate. Section 4 explores a variety of these values, i.e. a variety of endemic diseases.

- For the sentence “These go beyond current models...occur on them.”, I would recommend adding on the end of the sentence “to allow for modelling of a purposed future set of trades” to be more explicit to readers not well acquainted with related literature that this is the key component of the novelty of this paper.

Intro - For the 2nd paragraph, while it is stated that epidemic diseases like FMD are driven by livestock shipments, for endemic diseases, movements are unmentioned and that there are multiple transmission pathways with little understanding. As the model is focused on endemic diseases, I think a statement indicating that livestock movements have driven endemicity in other countries would be beneficial (along with any other evidence that livestock movement matters to endemicity). Here is a useful source for that: <https://pubmed.ncbi.nlm.nih.gov/15962561/>
- I think the sentence “With application to the Scottish cattle...” should be moved later in the paragraph, as a range of endemic diseases is explored on section 4, not section 3. I also believe this would be beneficial since on line 100, “Extending this to account for disease transmission via trade” inferred to me that disease dynamics would not be explored on a real world trading system before reading the section 4 myself.

Materials and Methods - In the first paragraph, I think explicitly defining that trades of farm i mean flow coming into i from other farms would be beneficial. The terminology “ i trades with j ” makes me think i shipped cattle to j , when the opposite is occurring.

- Table 1: For p_{ij} , the definition states that the parameter is the “probability that a trading partnership between i and j is present”. This infers to me that p_{ij} and p_{ji} are equal, since that definition does not indicate direction of trades. As it is possible for p_{ij} to not be equal to p_{ji} , I would reword the definition to “probability that a trading partnership made by i with j is present”.

- Line 119: The sentence states “annual average in- and out-flows of animals measure farm-level demand”, but in table 1, the same variables have the definition “annual in-flow referring to the same variable”. I would be consistent and have annual or annual average in both cases, as most will refer to the table. Similar point with line 169, as “per unit time” is used to refer to the same parameter of in-flow instead of being consistent with the previous text.

Results - Line 264: Is this reference correct? The paper doesn’t explain R_0 of a static network.

Case Study - I was curious why a removal rate of 0.2 was chosen for figure 2. If the general results for figure 2 are invariant of removal rate chosen (which I assume so), then I recommend stating this. If a removal rate was based on any endemic disease, I would recommend a citation as I do not know a livestock disease that would last on average 5 years on a farm.

- I think the bottom two plots of Figure 13 in the appendix should be in the main text as well to show all aspects of data fitting.

- Fig 3: In figure 4’s description, it is stated “The new trading patterns are those shown in Fig 3d”. Figure 3 does not have its plots labelled as a,b,c,d, so this should be included.

Discussion/Conclusion - I would insert somewhere that different network structures can have a different effect on which method of influencing partnerships (batch size, number of partnerships on average etc) is most effective. This is important to note, but not a major enough aspect to provide in the abstract.

Abstract - A3.2: For R_0 to differ so much on the homogeneous system is very unusual. Is there any reason why this could have happened? If simulations fared more accurately to theoretical results for the data example when reducing partnership duration, then perhaps note this in the text, as the behaviour could be due to the network topology.

- For A2, the limit for R_0 as $N \rightarrow \infty$ I could not derive with the information provided. I suggest to provide more steps for that calculation.

- In A4, for the first equation for $a_i(t)$, refer to the subscript of the sum. I know what was meant to be communicated was for the sum to not contain any j ’s which were previous partners. However, $k_{ij}^{in}(t-1)$ is the “Expected number of concurrent trading partners for farm i conditioned on zero partnerships at $t=0$ ”. This is not the set of previous partners i had, so the notation needs to be changed.

- Figure 9 has the x axis incorrectly labelled on the first plot, should be “Average size of trade”.

- Figures 10 and 12 have slightly different y axis labels when they are equivalent plots with different rates. Change accordingly.

Nitpicks

General - It was mentioned that Gillespie SSA was used to run simulations in the Appendix, but not the main text when one figure used a simulation. As it is required for the main text to stand on its own, this should be added into the description of figure 1.

- There is no section 4 on the paper, the sections go 1,2,3,5,6.

- Various inconsistencies in the references. For example, reference 6 does not have a page number (454), reference 8 states page number as 169-77 instead of 169-177, and references 18 and 19 have unnecessary capitalisation.

Materials and Methods - Table 1: β_{ij} should be equal to $\phi_{ij}B(\theta_{ij})$, not $\phi_{ij}B(\theta_{ij})$.

- Line 172: The sentence “This expression is easily interpreted...in a unit of time” Should indicate direction, i.e. trades from j to i .

Results - Line 315: Maybe provide some examples of markets playing a strong role in epidemics like in FMD?

- Lines 247-251: I think this part can be reworded to communicate the key point more clearly. Add "...so for all i , thus increasing the batch size reduces R_0 . In the case of..."
- Line 231: Reference (4) should be at the end of the sentence.
- Line 263: Equation is not separate from the rest of the paragraph.

Case Study - I found figure 4 confusing. I suggest to rename the x axis label as "compliance percentage of most frequent buyers".

- For figure 4, it is stated in the main text that "Case 1) 8% compliance is sufficient to bring R_0 below 1". This is not evident by looking at figure 4. Maybe it could be a good idea to add the initial R_0 values for all three cases in the description.
- Fig 1's description I would recommend highlighting Fig 13 in the Appendix, as I found it a useful visualisation.
- Line 343: Eq not on a separate line.
- Line 361: System average R_0 is ambiguous, I'd say R_0 is the system average of $R_{0\{i\}}$.
- Fig 3: The 4th plot needs an extra space in the title to separate two words.

Appendix - Figure 5 shows that on the whole network scale, the number of shipments and cattle is consistent each year to take an average. However, it is not explicitly stated or shown that this is true on a farm to farm basis to insure the averages used in the model are reasonable. For example, one year there could be a cluster of farms contributing large in/out-flow or shipments, while in other years, other locations will due to the changes in the industry. Individual farm contributions are likely consistent year to year, but the plot does not answer this directly. Maybe provide a source or give a comment stating that the industry hasn't changed much in Scotland, so individual farm contributions are consistent as well.

- In A4, the definition of $A_i(t-1,t)$ should state "the number of observed trading partners formed by farm i in year t ..." to explicitly incorporate time in the definition.
- For A2, I would choose another notation for probability of i infecting j . It is too similar to the probability of a partnership.

Decision letter (RSOS-201715.R0)

Dear Mr Knight

On behalf of the Editors, we are pleased to inform you that your Manuscript RSOS-201715 "Generative models of network dynamics provide insight into the effects of trade on endemic livestock disease" has been accepted for publication in Royal Society Open Science subject to minor revision in accordance with the referees' reports. Please find the referees' comments along with any feedback from the Editors below my signature.

Please submit your revised manuscript and required files (see below) no later than 7 days from today's (ie 25-Jan-2021) date. Note: the ScholarOne system will 'lock' if submission of the revision

is attempted 7 or more days after the deadline. If you do not think you will be able to meet this deadline please contact the editorial office immediately.

on behalf of Professor Joshua Ross (Associate Editor) and Mark Chaplain (Subject Editor)
openscience@royalsociety.org

Reviewer comments to Author:

Reviewer: 1
Comments to the Author(s)

I enjoyed reading this paper, finding it well written and easy to follow. I only have 2 comments that are not all that substantive:

1) In figure 2, the contrast showing the reduction in R_0 is difficult to see. I would suggest using a different colour map rather than shades of a single colour.

2) I have one comment on the units of some of the variables used in this paper: The rate at which farms form/dissolve partnerships is given in terms of inflows and outflows of animals. The units of these rates must be the number of partnerships formed/dissolved per unit time but the formation rate is proportional to η times ζ while the dissolution rate is inversely proportional to this product so I cannot see how the units are consistent unless η , ζ are dimensionless, meaning the coefficients a , d have units of 'per time'. Please can you explain.

Reviewer: 2
Comments to the Author(s)

Summary of Paper

The paper introduces a mechanistic model which considers partnership dynamics and trading behaviour within livestock to evaluate their effect on mitigating endemic disease. As the previous state of the art paper only considered the size and frequency of trades on a static network, this model is a new contribution to the literature. R_0 was quantified on a system and individual farm basis, affected by changes in partnership and trading dynamics. The effectiveness of specific partnership and trade changes can be affected by network topology. Using a case study of the Cattle Trading System data in Scotland: fewer, long-lasting trade partnerships and small amounts of large trades can reduce R_0 without reducing the demand of cattle on the farm-scale. These

results are encouraging, as these measures may have more of an effect on disease reduction than biosecurity increase and be less intrusive. Incentivising high-risk farms to adopt such changes introduces most of the effect of changing trade and partnerships on the entire network.

Recommendation/Overall Thoughts of Paper

Overall, I recommend this paper to be accepted after a few minor revisions. The intro sets an appropriate background for the reader, summarises recent research, and the novel aspect of the proposed model well. The methodology and derivations are largely sound, with sensible inferences being made from the results in a clearly presented manner. Further work and uncertainties of applying such measures to reduce R_0 in practise are well addressed. My revisions in the comments section are primarily for clarity rather than methodological issues.

However, I have one methodological aspect to note regarding the assumption of treating shipments via markets as transitory. The sources which justify this assumption on line 314 only justify not having markets as nodes for slow-spreading endemic diseases. For faster spreading endemic diseases like foot and mouth disease (FMD) for various countries, transmission can occur within a short period and can influence overall prevalence. I would either cite sources which provide evidence of markets contributing a small role in transmission for fast spread endemic diseases, or clarify that the range of endemic diseases modelled can only be slow-spreading within the text.

Minor Comments

Abstract - I would recommend moving "for a broad range of endemic diseases" (line 22) to the end of the next sentence. This is because "We show how these novel modelling tools enable systems approaches to disease control, using R_0 to explore impacts of changes in trading practices on between-farm prevalence levels" refers to the work in section 3, which was only done for one value for the on-farm prevalence and removal rate. Section 4 explores a variety of these values, i.e. a variety of endemic diseases.

- For the sentence "These go beyond current models...occur on them.", I would recommend adding on the end of the sentence "to allow for modelling of a purposed future set of trades" to be more explicit to readers not well acquainted with related literature that this is the key component of the novelty of this paper.

Intro - For the 2nd paragraph, while it is stated that epidemic diseases like FMD are driven by livestock shipments, for endemic diseases, movements are unmentioned and that there are multiple transmission pathways with little understanding. As the model is focused on endemic diseases, I think a statement indicating that livestock movements have driven endemicity in other countries would be beneficial (along with any other evidence that livestock movement matters to endemicity). Here is a useful source for that: <https://pubmed.ncbi.nlm.nih.gov/15962561/>
- I think the sentence "With application to the Scottish cattle..." should be moved later in the paragraph, as a range of endemic diseases is explored on section 4, not section 3. I also believe this would be beneficial since on line 100, "Extending this to account for disease transmission via trade" inferred to me that disease dynamics would not be explored on a real world trading system before reading the section 4 myself.

Materials and Methods - In the first paragraph, I think explicitly defining that trades of farm i mean flow coming into i from other farms would be beneficial. The terminology " i trades with j " makes me think i shipped cattle to j , when the opposite is occurring.

- Table 1: For p_{ij} , the definition states that the parameter is the "probability that a trading partnership between i and j is present". This infers to me that p_{ij} and p_{ji} are equal, since that definition does not indicate direction of trades. As it is possible for p_{ij} to not be equal to p_{ji} , I

would reword the definition to “probability that a trading partnership made by i with j is present”.

- Line 119: The sentence states “annual average in- and out-flows of animals measure farm-level demand”, but in table 1, the same variables have the definition “annual in-flow referring to the same variable”. I would be consistent and have annual or annual average in both cases, as most will refer to the table. Similar point with line 169, as “per unit time” is used to refer to the same parameter of in-flow instead of being consistent with the previous text.

Results - Line 264: Is this reference correct? The paper doesn’t explain R_0 of a static network.

Case Study - I was curious why a removal rate of 0.2 was chosen for figure 2. If the general results for figure 2 are invariant of removal rate chosen (which I assume so), then I recommend stating this. If a removal rate was based on any endemic disease, I would recommend a citation as I do not know a livestock disease that would last on average 5 years on a farm.

- I think the bottom two plots of Figure 13 in the appendix should be in the main text as well to show all aspects of data fitting.

- Fig 3: In figure 4’s description, it is stated “The new trading patterns are those shown in Fig 3d”. Figure 3 does not have its plots labelled as a,b,c,d, so this should be included.

Discussion/Conclusion - I would insert somewhere that different network structures can have a different effect on which method of influencing partnerships (batch size, number of partnerships on average etc) is most effective. This is important to note, but not a major enough aspect to provide in the abstract.

Abstract - A3.2: For R_0 to differ so much on the homogeneous system is very unusual. Is there any reason why this could have happened? If simulations fared more accurately to theoretical results for the data example when reducing partnership duration, then perhaps note this in the text, as the behaviour could be due to the network topology.

- For A2, the limit for R_0 as $N \rightarrow \infty$ I could not derive with the information provided. I suggest to provide more steps for that calculation.

- In A4, for the first equation for $a_i(t)$, refer to the subscript of the sum. I know what was meant to be communicated was for the sum to not contain any j ’s which were previous partners. However, $k_{ini}(t-1)$ is the “Expected number of concurrent trading partners for farm i conditioned on zero partnerships at $t=0$ ”. This is not the set of previous partners i had, so the notation needs to be changed.

- Figure 9 has the x axis incorrectly labelled on the first plot, should be “Average size of trade”.

- Figures 10 and 12 have slightly different y axis labels when they are equivalent plots with different rates. Change accordingly.

Nitpicks

General - It was mentioned that Gillespie SSA was used to run simulations in the Appendix, but not the main text when one figure used a simulation. As it is required for the main text to stand on its own, this should be added into the description of figure 1.

- There is no section 4 on the paper, the sections go 1,2,3,5,6.

- Various inconsistencies in the references. For example, reference 6 does not have a page number (454), reference 8 states page number as 169-77 instead of 169-177, and references 18 and 19 have unnecessary capitalisation.

Materials and Methods - Table 1: β_{ij} should be equal to $\phi_{ji}B(\theta_j)$, not $\phi_{ij}B(\theta_j)$.

- Line 172: The sentence “This expression is easily interpreted...in a unit of time” Should indicate direction, i.e. trades from j to i .

Results - Line 315: Maybe provide some examples of markets playing a strong role in epidemics like in FMD?

- Lines 247-251: I think this part can be reworded to communicate the key point more clearly. Add "...so for all i , thus increasing the batch size reduces R_0 . In the case of..."
- Line 231: Reference (4) should be at the end of the sentence.
- Line 263: Equation is not separate from the rest of the paragraph.

Case Study - I found figure 4 confusing. I suggest to rename the x axis label as "compliance percentage of most frequent buyers".

- For figure 4, it is stated in the main text that "Case 1) 8% compliance is sufficient to bring R_0 below 1". This is not evident by looking at figure 4. Maybe it could be a good idea to add the initial R_0 values for all three cases in the description.
- Fig 1's description I would recommend highlighting Fig 13 in the Appendix, as I found it a useful visualisation.
- Line 343: Eq not on a separate line.
- Line 361: System average R_0 is ambiguous, I'd say R_0 is the system average of $R_{0\{i\}}$.
- Fig 3: The 4th plot needs an extra space in the title to separate two words.

Appendix - Figure 5 shows that on the whole network scale, the number of shipments and cattle is consistent each year to take an average. However, it is not explicitly stated or shown that this is true on a farm to farm basis to insure the averages used in the model are reasonable. For example, one year there could be a cluster of farms contributing large in/out-flow or shipments, while in other years, other locations will due to the changes in the industry. Individual farm contributions are likely consistent year to year, but the plot does not answer this directly. Maybe provide a source or give a comment stating that the industry hasn't changed much in Scotland, so individual farm contributions are consistent as well.

- In A4, the definition of $A_i(t-1,t)$ should state "the number of observed trading partners formed by farm i in year t ..." to explicitly incorporate time in the definition.
- For A2, I would choose another notation for probability of i infecting j . It is too similar to the probability of a partnership.

===PREPARING YOUR MANUSCRIPT===

===PREPARING YOUR REVISION IN SCHOLARONE===

Author's Response to Decision Letter for (RSOS-201715.R0)

See Appendix A.

Decision letter (RSOS-201715.R1)

Dear Mr Knight,

It is a pleasure to accept your manuscript entitled "Generative models of network dynamics provide insight into the effects of trade on endemic livestock disease" in its current form for publication in Royal Society Open Science.

on behalf of Professor Joshua Ross (Associate Editor) and Mark Chaplain (Subject Editor)
openscience@royalsociety.org

Appendix A

Reviewer 1

- 1) In figure 2, the contrast showing the reduction in R_0 is difficult to see. I would suggest using a different colour map rather than shades of a single colour.

Accepted and colour amended.

- 2) I have one comment on the units of some of the variables used in this paper: The rate at which farms form/dissolve partnerships is given in terms of inflows and outflows of animals. The units of these rates must be the number of partnerships formed/dissolved per unit time but the formation rate is proportional to η times ζ while the dissolution rate is inversely proportional to this product so I cannot see how the units are consistent unless η, ζ are dimensionless, meaning the coefficients a, d have units of 'per time'. Please can you explain.
You are correct that if η and ζ have dimensions t^{-1} then the formation and cessation rates are inconsistent. However, we take η and ζ to be dimensionless quantities as our chosen timescales over which they are averaged is arbitrary, e.g. we could choose to take them as per 2 year averages. We can view η and ζ as a measurement of the size of the farm and interpret the constants a, d , and b as being rates with dimension t^{-1} so that a consistent dimension is maintained for the partnership formation and cessations rates, and the trade rate. Table 1 has been amended to reflect these changes.

Reviewer 2

I have one methodological aspect to note regarding the assumption of treating shipments via markets as transitory. The sources which justify this assumption on line 314 only justify not having markets as nodes for slow-spreading endemic diseases. For faster spreading endemic diseases like foot and mouth disease (FMD) for various countries, transmission can occur within a short period and can influence overall prevalence. I would either cite sources which provide evidence of markets contributing a small role in transmission for fast spread endemic diseases, or clarify that the range of endemic diseases modelled can only be slow-spreading within the text.
The implication is that we are considering only diseases for which transmission is slow so that our assumption that markets movements do not significantly affect disease spread is not violated. We concede, however, that some of our chosen disease parameterisations in Figure 3 may represent fast spreading endemic diseases, in particular those with high on-farm prevalence and short infectious periods. In these cases, markets may have a more prominent role in the transmission of disease, and the inclusion of market-like nodes in our model is an extension of our work that we hope to consider in the future.

Abstract - I would recommend moving “for a broad range of endemic diseases” (line 22) to the end of the next sentence. This is because “We show how these novel modelling tools enable systems approaches to disease control, using R_0 to explore impacts of changes in trading practices on between-farm prevalence levels” refers to the work in section 3, which was only done for one value for the on-farm prevalence and removal rate. Section 4 explores a variety of these values, i.e. a variety of endemic diseases. ***Accepted***

For the sentence “These go beyond current models...occur on them.”, I would recommend adding on the end of the sentence “to allow for modelling of a purposed future set of trades” to be more explicit to readers not well acquainted with related literature that this is the key component of the novelty of

this paper. **Declined. The novelty is in the explicit modelling of the dynamics of trade partnerships, which is highlighted in this sentence. Furthermore, "...purposed future set of trades" may be interpreted to represent a fixed set of trades, which our model neither requires nor enforces.**

Intro - For the 2nd paragraph, while it is stated that epidemic diseases like FMD are driven by livestock shipments, for endemic diseases, movements are unmentioned and that there are multiple transmission pathways with little understanding. As the model is focused on endemic diseases, I think a statement indicating that livestock movements have driven endemicity in other countries would be beneficial (along with any other evidence that livestock movement matters to endemicity). Here is a useful source for that <https://pubmed.ncbi.nlm.nih.gov/15962561/> **Accepted. First sentence of introduction provides references to disease spread in livestock networks being driven by animal movements, however a comment explicitly highlighting animal movements as a factor in the challenge of controlling endemic diseases has been added.**

I think the sentence "With application to the Scottish cattle..." should be moved later in the paragraph, as a range of endemic diseases is explored on section 4, not section 3. I also believe this would be beneficial since on line 100, "Extending this to account for disease transmission via trade" inferred to me that disease dynamics would not be explored on a real world trading system before reading the section 4 myself. **Accepted**

Materials and Methods - In the first paragraph, I think explicitly defining that trades of farm i mean flow coming into i from other farms would be beneficial. The terminology " i trades with j " makes me think i shipped cattle to j , when the opposite is occurring. **Accepted**

Table 1: For p_{ij} , the definition states that the parameter is the "probability that a trading partnership between i and j is present". This infers to me that p_{ij} and p_{ji} are equal, since that definition does not indicate direction of trades. As it is possible for p_{ij} to not be equal to p_{ji} , I would reword the definition to "probability that a trading partnership made by i with j is present". **Accepted**

Line 119: The sentence states "annual average in- and out-flows of animals measure farm-level demand", but in table 1, the same variables have the definition "annual in-flow referring to the same variable". I would be consistent and have annual or annual average in both cases, as most will refer to the table. Similar point with line 169, as "per unit time" is used to refer to the same parameter of in-flow instead of being consistent with the previous text. **Accepted**

Results - Line 264: Is this reference correct? The paper doesn't explain R_0 of a static network. **The reference derives an expression for R_0 without explicitly stating that the result is equivalent to R_0 .**

Case Study - I was curious why a removal rate of 0.2 was chosen for figure 2. If the general results for figure 2 are invariant of removal rate chosen (which I assume so), then I recommend stating this. If a removal rate was based on any endemic disease, I would recommend a citation as I do not know a livestock disease that would last on average 5 years on a farm. **The choice of removal rate was not intended to represent any particular endemic disease, rather it was used to show the potential of our methods for a disease that is highly persistent. However, paraTB is known to persist for significant lengths of time on farms, due to its ability to persist for 1 year in farm soil, poor diagnostic tests, and farmers' perception of the disease resulting in poor biosurveillance.**

I think the bottom two plots of Figure 13 in the appendix should be in the main text as well to show all aspects of data fitting. **Declined due to space concerns.**

Fig 3: In figure 4's description, it is stated "The new trading patterns are those shown in Fig 3d". Figure 3 does not have its plots labelled as a, b, c, d, so this should be included. **Accepted**

Discussion/Conclusion - I would insert somewhere that different network structures can have a different effect on which method of influencing partnerships (batch size, number of partnerships on average etc) is most effective. This is important to note, but not a major enough aspect to provide in the abstract. **Accepted**

Abstract - A3.2: For R_0 to differ so much on the homogeneous system is very unusual. Is there any reason why this could have happened? If simulations fared more accurately to theoretical results for the data example when reducing partnership duration, then perhaps note this in the text, as the behaviour could be due to the network topology. **We suspect these differences are due to network topology and edge density (which, at 0.01, is quite high compared to most real-world networks), and potential modelling assumptions made to derive R_0 , in particular that the network is approximately tree-like. For high density networks where edges become more permanent, the probability of loops in the network increases which impact disease transmission.**

For A2, the limit for R_0^i as $N \rightarrow \infty$ I could not derive with the information provided. I suggest to provide more steps for that calculation. **Accepted. More description given**

In A4, for the first equation for $a_i(t)$, refer to the subscript of the sum. I know what was meant to be communicated was for the sum to not contain any j 's which were previous partners. However, $k_i^{in}(t-1)$ is the "Expected number of concurrent trading partners for farm i conditioned on zero partnerships at $t=0$ ". This is not the set of previous partners i had, so the notation needs to be changed. **Accepted.**

Figure 9 has the x axis incorrectly labelled on the first plot, should be "Average size of trade". **Accepted**

Figures 10 and 12 have slightly different y axis labels when they are equivalent plots with different rates. Change accordingly. **Accepted**

Nitpicks

General - It was mentioned that Gillespie SSA was used to run simulations in the Appendix, but not the main text when one figure used a simulation. As it is required for the main text to stand on its own, this should be added into the description of figure 1. **Accepted**

There is no section 4 on the paper, the sections go 1,2,3,5,6. **Corrected when reformatting paper**

Various inconsistencies in the references. For example, reference 6 does not have a page number (454), reference 8 states page number as 169–77 instead of 169–177, and references 18 and 19 have unnecessary capitalisation. **Accepted**

Materials and Methods - Table 1: β_{ij} should be equal to $\phi_{ij}B(\theta_j)$, not $\phi_{ij}B(\theta_i)$. **Accepted**

Line 172: The sentence “This expression is easily interpreted...in a unit of time” Should indicate direction, i.e. trades from j to i . **Accepted**

Results - Line 315: Maybe provide some examples of markets playing a strong role in epidemics like in FMD? **Accepted**

Lines 247-251: I think this part can be reworded to communicate the key point more clearly. Add “...so for all i , thus increasing the batch size reduces R_0 . In the case of...”. **Accepted**

Line 231: Reference (4) should be at the end of the sentence. **This is a reference to Eq. (4), not a literature reference.**

Line 263: Equation is not separate from the rest of the paragraph. **This was a formatting error during submission. Corrected in final version**

Case Study - I found figure 4 confusing. I suggest to rename the x axis label as “compliance percentage of most frequent buyers”. **Accepted**

For figure 4, it is stated in the main text that “Case 1) 8% compliance is sufficient to bring R_0 below 1”. This is not evident by looking at figure 4. Maybe it could be a good idea to add the initial R_0 values for all three cases in the description. **Accepted**

Fig 1’s description I would recommend highlighting Fig 13 in the Appendix, as I found it a useful visualisation. **Accepted**

Line 343: Eq not on a separate line. **Formatting error on initial submission. Corrected in final version**

Line 361: System average R_0 is ambiguous, I’d say R_0 is the system average of R_{0_i} . **Accepted**

Fig 3: The 4th plot needs an extra space in the title to separate two words. **Accepted**

Appendix - Figure 5 shows that on the whole network scale, the number of shipments and cattle is consistent each year to take an average. However, it is not explicitly stated or shown that this is true on a farm to farm basis to insure the averages used in the model are reasonable. For example, one year there could be a cluster of farms contributing large in/out-flow or shipments, while in other years, other locations will due to the changes in the industry. Individual farm contributions are likely consistent year to year, but the plot does not answer this directly. Maybe provide a source or give a comment stating that the industry hasn’t changed much in Scotland, so individual farm contributions are consistent as well. **Accepted. Comment added with reference**

In A4, the definition of $A_i(t-1,t)$ should state “the number of observed trading partners formed by farm i in year t ...” to explicitly incorporate time in the definition. **Accepted**

For A2, I would choose another notation for probability of i infecting j . It is too similar to the probability of a partnership. **Accepted**